# Emerging Role of ABC Transporters in Glia Cells in Health and Diseases of the Central Nervous System

**DOI:** 10.3390/cells13090740

**Published:** 2024-04-24

**Authors:** Maria Villa, Jingyun Wu, Stefanie Hansen, Jens Pahnke

**Affiliations:** 1Translational Neurodegeneration Research and Neuropathology Lab, Department of Clinical Medicine (KlinMed), Medical Faculty, University of Oslo (UiO) and Section of Neuropathology Research, Department of Pathology (PAT), Clinics for Laboratory Medicine (KLM), Oslo University Hospital (OUS), Sognsvannsveien 20, NO-0372 Oslo, Norway; 2Institute of Nutritional Medicine (INUM)/Lübeck Institute of Dermatology (LIED), University of Lübeck (UzL) and University Medical Center Schleswig-Holstein (UKSH), Ratzeburger Allee 160, D-23538 Lübeck, Germany; 3Department of Pharmacology, Faculty of Medicine, University of Latvia (LU), Jelgavas iela 3, LV-1004 Rīga, Latvia; 4School of Neurobiology, Biochemistry and Biophysics, The Georg S. Wise Faculty of Life Sciences, Tel Aviv University (TAU), Tel Aviv IL-6997801, Israel

**Keywords:** ABCA1, ABCA7, ABCB1, ABCC1, MDR1, MRP1, P-gp, ABC transporter, neurodegeneration, glia, astrocyte, oligodendrocyte, microglia, multiple sclerosis, Alzheimer’s disease, Huntington’s disease, neuroinflammation, demyelination, metabolic diseases, rare diseases, EAE, steroid hormones

## Abstract

ATP-binding cassette (ABC) transporters play a crucial role for the efflux of a wide range of substrates across different cellular membranes. In the central nervous system (CNS), ABC transporters have recently gathered significant attention due to their pivotal involvement in brain physiology and neurodegenerative disorders, such as *Alzheimer’s disease* (AD). Glial cells are fundamental for normal CNS function and engage with several ABC transporters in different ways. Here, we specifically highlight ABC transporters involved in the maintenance of brain homeostasis and their implications in its metabolic regulation. We also show new aspects related to ABC transporter function found in less recognized diseases, such as *Huntington’s disease* (HD) and *experimental autoimmune encephalomyelitis* (EAE), as a model for *multiple sclerosis* (MS). Understanding both their impact on the physiological regulation of the CNS and their roles in brain diseases holds promise for uncovering new therapeutic options. Further investigations and preclinical studies are warranted to elucidate the complex interplay between glial ABC transporters and physiological brain functions, potentially leading to effective therapeutic interventions also for rare CNS disorders.

## Table of Contents

**1. Introduction**…………………………………………………………………………….2* 1.1. ABC Transporters in Physiological Brain Function*………………………………..3  1.1.1. ABC Transporters and Neuronal Lipid Metabolism
  1.1.2. ABC Transporters and Neuronal Drug Efflux
* 1.2. Oligodendrocytes*………………………………………………………………….4  1.2.1. ABC Transporters in Oligodendrocyte Differentiation and Myelination
  1.2.2. ABC Transporters in Oligodendrocyte Survival and Protection
  1.2.3. ABC Transporters in Oligodendrocyte-Mediated Ion Homeostasis
* 1.3. Astrocytes*…………………………………………………………………………5  1.3.1. ABC Transporters in Astrocyte–Neuron Communication
  1.3.2. ABC Transporters in Astrocyte-Mediated Blood–Brain Barrier Function
* 1.4. ABC Transporters in Blood–Brain Barrier Integrity*…………………………….6  1.4.1. ABC Transporters in Nutrient Transport across the BBB
  1.4.2. ABC Transporters in Neurovascular Coupling
* 1.5. ABC Transporter Function in Immune Cells of the Brain*…………………………8  1.5.1. Lymphocytes/Monocytes
  1.5.2. Microglia
  1.5.3. Macrophages
**2. Function of ABC Transporters in Brain Diseases**……………………………………13* 2.1. ABC Transporter Dysfunction in Neurodegenerative Disorders*…………………13  2.1.1. ABC Transporters as Risk Factor for Alzheimer’s Disease
  2.1.2. ABC Transporters in Movement Disorders
* 2.2. Other, Non-Inflammatory Brain Diseases*………………………………………..17  2.2.1. White Matter Disorders—*Adrenoleukodystrophy* ABCD1
  2.2.2. ABC Transporters in Epilepsy
* 2.3. ABC Transporter Function in Neuroinflammatory Diseases*………………………18  2.3.1. Multiple Sclerosis
  2.3.2. Detecting Clinical Effects of ABC Transporter-deficiency in EAE
**3. General Physiological Effects of ABC Transporters on Modifying Diseases** ….....21* 3.1. Export of Toxic Metabolites/Peptides*………………………………………………21* 3.2. Examples of Molecular Mechanisms Influenced by ABC Transporters*……………21  3.2.1. Modulation of Steroid Hormone Signaling
**4. Summary**………………………………………………………………………………….23**References**………………………………………………………………………………...24

## 1. Introduction

Neurological diseases encompass a wide range of disorders that affect various functions of the central nervous system (CNS), both grossly or subtly, and often cell-specifically, leading to impairment in motor control and co-ordination, and memory and cognition deficiencies, as well as other neurological symptoms [1,2]. Amongst various molecular mechanisms implicated in the pathogenesis of neurological diseases, the role of ATP-binding cassette (ABC) transporters has gained increasing attention during the past two decades [3,4,5]. ABC transporters are a superfamily of transmembrane proteins that play essential roles in the transportation of a diverse range of molecules across cellular membranes. The expression and activity of ABC transporters in the CNS are critical for normal brain function. Specifically, these transporters are expressed at the blood–brain (BBB) and blood–choroid plexus barriers (BCPB), where they control the influx and efflux of endogenous compounds, xenobiotics, and therapeutic drugs [6,7]. They are involved in maintaining the brain’s homeostasis, regulating the entry and efflux of substrates, and protecting cells against toxic compounds [4,8]. ABC transporters are differentially expressed in different brain regions [9,10] and cell types, including neurons and glia cells such as astrocytes and oligodendrocytes, in which they play essential roles, e.g., for maintaining cellular lipid homeostasis, regulating neurotransmitter levels, and modulating inflammatory responses in the CNS [6,7].

Emerging evidence suggests that dysfunction of ABC transporters contributes to the development and progression of various neurological and neurodegenerative diseases, including Alzheimer’s disease (AD) [11,12,13,14], Parkinson’s disease (PD) [15,16,17,18,19], Huntington’s disease (HD) [20,21], and drug-resistant epilepsy [22,23,24,25,26]. For instance, in AD, alterations of ABC transporters such as ABCA1, ABCA7, ABCB1, ABCC1, and ABCG2 have been associated with the impaired clearance of amyloid-β (Aβ) peptides, leading to their accumulation in the brain and vessel walls and the formation of neurotoxic plaques or cerebral amyloid angiopathy (CAA) [11,12,13,14,27,28,29,30].

Understanding the precise roles of ABC transporters in neurological and neurodegenerative diseases is of great importance for several reasons. Firstly, elucidating the mechanisms underlying their dysregulation can provide valuable insights into disease pathogenesis and identify potential therapeutic targets. Secondly, ABC transporters have the potential to influence the pharmacokinetics and efficacy of therapeutic drugs used for the treatment of neurological diseases [7]. Thirdly, studying the interplay between ABC transporters and other tissue and cellular processes, such as inflammation and oxidative stress, can help to unravel complex disease mechanisms [5].

In this review, we will summarize the current knowledge about the involvement of ABC transporters in neurological and neurodegenerative diseases with a special emphasis on glial cells. We will discuss their expression patterns, physiological functions, and emerging evidence implicating their dysfunction in disease pathogenesis, and highlight their significance for the development of effective therapeutic interventions.

### 1.1. ABC Transporters in Physiological Brain Function

Neurons are highly specialized cells responsible for maintaining brain homeostasis and generating and/or transmitting electrical signals in the brain [31,32]. Besides basic metabolic mechanisms, the proper functioning of neurons relies on the maintenance of precise intra- and extracellular ion concentrations [33,34], and the synthesis and reuptake of neurotransmitters [35] in different cellular compartments and in other cells, such as astrocytes [36], as well as on the lipid composition of cellular and intracellular membranes [37,38].

In this regard, ABC transporters have been recognized as key players for maintaining neuronal homeostasis by regulating the transport of various substrates across the neuronal cell membrane (Figure 1). These transporters are located in the cell membranes of various glial cells, such as astrocytes, microglia, and oligodendrocytes, and are crucial for maintaining homeostasis of the brain microenvironment. For example, ABC transporters participate in the regulation of neurotransmitter levels by controlling their reuptake and recycling [39,40]. In addition, ABC transporters are involved in the transport of lipids and lipoproteins. Thereby, they contribute to sustaining the structural integrity and dynamic composition of cellular membranes [41,42,43]. Moreover, ABC transporters are crucial in transporting various metabolites, such as sugars, amino acids, lipids, and other small molecules involved in energy production and cellular homeostasis, ensuring the right balance for metabolic processes within neurons [44,45,46,47,48].

#### 1.1.1. ABC Transporters and Neuronal Lipid Metabolism

The lipid metabolism plays a fundamental role in neuronal structure, function, and signaling [49]. ABC transporters play a vital role in this metabolism as they constitute a significant family of proteins that contribute to neuronal lipid homeostasis by regulating the translocation of lipids across neuronal membranes [50,51]. ABCA1, ABCA2, ABCA7, and ABCG1 are integral components in the intricate regulation of phospholipid and cholesterol homeostasis, playing crucial roles in neuronal cholesterol efflux [52,53,54]. For instance, by regulating cholesterol efflux, they help maintain the cellular cholesterol balance and prevent the accumulation of cholesterol in neurons [21,55]. The association between cholesterol levels and Aβ peptide generation is a critical aspect of AD research. The involvement of ABCA1, ABCA7, and ABCG1 in modulating APP processing adds supportive evidence to these transporters constituting potential therapeutic targets for mitigating Aβ production [52,56]. ABCG1 and ABCG4 transporters have also been identified as key regulators of cellular lipid transport in neurons, affecting the composition of lipid rafts and the processing of amyloid-β precursor protein (APP) [57,58,59]. Moreover, ABCD1, ABCD2, and ABCD3 have been implicated in the transport of fatty acids into peroxisomes, underscoring their role in lipid catabolism and the etiology of peroxisomal disorders [60]. Recent research suggested that the expression of the ABCC4 transporter influences the accumulation of bioactive lipids, such as eicosanoids, which are associated with inflammation and neuronal signaling [61,62]. Overall, the intricate interplay between ABC transporters and neuronal lipid metabolism highlights their crucial role in maintaining neuronal membrane integrity, synaptic function, and overall brain health.

#### 1.1.2. ABC Transporters and Neuronal Drug Efflux

Neurons are often exposed to various exogenous compounds, including therapeutic drugs, environmental toxins, and xenobiotics. ABC transporters, particularly ABCB1, also known as P-glycoprotein (P-gp), and ABCG2, also known as breast cancer resistance protein (BCRP), are vital components of the neuronal drug efflux mechanism. They hereby play a crucial role in protecting neurons from the potential toxicity of these compounds by modulating the pharmacokinetic and pharmacodynamic profiles of therapeutic agents and actively transporting them out of the cells [26,63,64].

ABCB1 is highly expressed in neurons and has been extensively studied for its efflux functions [65]. This transporter can limit the entry of drugs into neurons and contribute to the development of drug resistance in neurological disorders such as Amyotrophic Lateral Sclerosis (ALS). In ALS, an induced pharmaco-resistance has been observed, suggesting a selective upregulation of these transporters in disease-affected CNS regions, both at the mRNA and protein levels [64,66,67,68]. In epilepsy, the overexpression of ABCB1 is proven to be an important factor for the limited efficacy of antiepileptic drugs, as these transporters reduce the effective drug concentration at the site of lesions [69,70]. Moreover, the expression of other ABC transporters, such as ABCG2 and ABCC1, has been detected in neurons and implicated in the efflux of drugs and toxic metabolites [63]. Thus, understanding the mechanisms of neuronal drug efflux mediated by ABC transporters is essential for optimizing drug delivery to the brain and developing strategies to overcome drug resistance in the treatment of neurological conditions [71].

### 1.2. Oligodendrocytes

#### 1.2.1. ABC Transporters in Oligodendrocyte Differentiation and Myelination

Oligodendrocytes are glial cells in the CNS that play a crucial role for myelination, the process of forming the myelin sheath around neuronal axons and dendrites [72,73]. Myelin provides insulation and facilitates efficient electrical signal transmission [73]. Recent studies have proven that ABC transporters are implicated in the complex processes of oligodendrocyte differentiation and myelination. For instance, ABCA2 has been identified as a key transporter involved in lipid trafficking and synthesis during myelination, with its expression predominantly occurring in oligodendrocytes during the maturation of the developing rat brain [3,74]. This evidence implies that ABCA2 plays a crucial role in the transport of cholesterol and other lipids to the developing myelin sheaths [74,75]. Furthermore, ABC transporters such as ABCG1 and ABCG4 have been shown to regulate the efflux of lipids from oligodendrocytes, which, in turn, impacts myelin integrity and maintenance [76,77].

#### 1.2.2. ABC Transporters in Oligodendrocyte Survival and Protection

ABCC1, also known as multidrug resistance-associated protein 1 (MRP1), is highly expressed in oligodendrocytes and plays a crucial role in the export of potentially toxic metabolites and xenobiotics [78]. Dysfunction of ABCC1 has been associated with impaired oligodendrocyte survival and susceptibility to oxidative stress [79,80]. Additionally, another key transporter, ABCG2, has been suggested to contribute to cellular survival by protecting against apoptosis induced by various stressors, including cytokines. While this research is primarily focused on placental cells, the protective mechanisms of ABCG2 may be extrapolated to oligodendrocytes, where similar stress responses are crucial for cell survival [81]. Notably, ABC transporters such as ABCB1 and ABCG2 contribute to the efflux of neurotoxic compounds from oligodendrocytes, thereby protecting them from damage [28]. Nevertheless, it is important to consider that the precise mechanisms of ABC transporters in oligodendrocyte survival and protection are still being elucidated.

#### 1.2.3. ABC Transporters in Oligodendrocyte-Mediated Ion Homeostasis

Oligodendrocytes also play a critical role in maintaining ion homeostasis in the CNS, which is essential for proper neuronal function [73]. ABC transporters expressed in oligodendrocytes contribute to the regulation of ion concentrations in the extracellular environment [82]. For instance, the ATP-dependent ion transporter sulfonylurea receptor 1 (SUR1)/ABCC8 has been shown to be involved in potassium channel regulation in oligodendrocytes [83,84], preventing excessive potassium accumulation which could be detrimental to neuronal function. Furthermore, ABC transporters have been identified as crucial for maintaining metal ion homeostasis in microorganisms [85,86,87,88], which may provide insights into similar functions in oligodendrocytes. For example, the identified metal-specific ABC transport systems, while studied in bacteria, suggest a conserved mechanism that could be relevant for oligodendrocyte function, and, here particularly, for iron ion uptake, which is important for myelin production [89]. While little research specifically focusing on ABC transporters in oligodendrocyte-mediated ion homeostasis is available, it is worth emphasizing that the dysregulation of ion transporters in oligodendrocytes can lead to imbalances in ion concentrations for myelination processes, compromising neuronal excitability and neurotransmission [84].

In conclusion, ABC transporters expressed in oligodendrocytes play crucial roles in oligodendrocyte differentiation, myelination, survival, protection, and ion homeostasis (Figure 1). By regulating lipid transport, the efflux of toxic compounds, and ion balance, ABC transporters actively contribute to the physiologic function of oligodendrocytes and the maintenance of myelin integrity. Dysregulation or dysfunction of oligodendrocytic ABC transporters can have profound implications for myelin formation and function.

### 1.3. Astrocytes

#### 1.3.1. ABC Transporters in Astrocyte–Neuron Communication

Astrocytes are the most abundant glial cells in the brain, supporting neuronal function and maintaining brain homeostasis. One key aspect of astrocyte function is their participation in astrocyte–neuron communication, whereby they regulate the extracellular environment and provide essential metabolic support to neurons [90]. Details on ABC transporters in astrocyte–neuron communication are limited. However, several studies highlight the importance of glial cells and their interactions with neurons through various mechanisms, indirectly indicating the potential involvement of ABC transporters. For example, ABCC1 is involved in the efflux of glutathione, glutathione-conjugates, and other organic anions from astrocytes, which can modulate synaptic transmission and neuronal excitability [91,92]. Additionally, the SUR1/ABCC8 is highly expressed in astrocytes and regulates potassium fluxes, which influence neuronal excitability and synaptic activity [93,94]. Furthermore, ABCA1 and ABCG1 play pivotal roles in cholesterol efflux from astrocytes to neurons, contributing to the regulation of cholesterol homeostasis in the brain [48,95,96].

#### 1.3.2. ABC Transporters in Astrocyte-Mediated Blood–Brain Barrier Function

Astrocytes play a vital role in maintaining the integrity and function of the BBB, a highly selective barrier that serves as an interface between the CNS and the peripheral circulation and regulates the passage of molecules between the blood and the brain [90]. ABC transporters expressed in astrocyte processes contribute to the efflux of xenobiotics and drugs across the BBB, limiting their entry into the brain (Figure 1) [97]. The activity of these astrocytic ABC transporters can influence drug efficacy, brain drug concentrations, and susceptibility to neurotoxic compounds. For instance, the drug resistance transporters ABCB1 and ABCG2 are highly expressed in astrocyte end-feet where they actively transport a wide range of substances back into the bloodstream, including therapeutic drugs and toxins [98,99]. Interestingly, chronic intermittent hypoxia and sustained hypoxia can both lead to BBB dysfunction characterized by the altered expression and activity of ABCB1 and ABCG2 [100]. This finding highlights the impact of environmental factors on BBB integrity and its associated transport mechanisms. In terms of astrocyte-mediated functions, the astrocyte-derived apolipoprotein E4 (APOE4) was proven to impair BBB integrity, leading to increased leakage and decreased astrocyte end-feet coverage of blood vessels, suggesting a role for astrocyte-derived factors in BBB function and transporter expression [101].

Overall, ABC transporters expressed in astrocytes are integral to the dynamic communication between astrocytes and neurons (Figure 1). They play a pivotal role in the function of the BBB, where they control the transit of a variety of molecules, including neurotransmitters, ions, and xenobiotics. This function is essential not only for normal cerebral operations but also for generally preserving CNS equilibrium. Their involvement in drug resistance, particularly in relation to CNS pharmacotherapy, underscores their potential as therapeutic targets to enhance drug delivery to the brain and address CNS pathologies [100,102]. Therefore, astrocytic ABC transporters are not only essential for maintaining a homeostatic environment conducive to optimal neuronal function, but also offer a promising avenue for therapeutic intervention.

### 1.4. ABC Transporters in Blood–Brain Barrier Integrity

The BBB is a highly selective and dynamic structure composed of endothelial cells, pericytes, and astrocyte end-feet that regulates the exchange of substances between the blood and the brain [103]. ABC transporters expressed in endothelial cells and pericytes play a crucial role in maintaining BBB integrity (Figure 1) [63]. Among other ABC transporters, ABCB1 and ABCG2 are highly expressed in endothelial cells of the BBB, where they actively pump out a wide range of substances from the brain, including toxic compounds and therapeutic drugs [104,105]. This protective function is vital, not only for normal brain physiology but also for minimizing the impact of age-related neurodegenerative diseases on BBB integrity [106]. However, the expression of ABC transporters can be modulated by various factors, including oxidative stress, diet and environmental pollutants. With regard to BBB integrity, this modulation has implications for drug delivery to the brain. It can be beneficial in some cases, but also presents challenges for the treatment of CNS disorders. Innovative approaches are being explored to target signaling events that could reduce the activity of ABC transporters, thereby enhancing drug delivery to the brain [107]. Additionally, pericytes express various ABC transporters, such as ABCC1, which contributes to the efflux of xenobiotics and metabolites across the BBB [10]. The co-ordinated action of ABC transporters in endothelial cells and pericytes is essential for maintaining the selective permeability of the BBB and preventing the entry of neurotoxic substances into the brain [108].

#### 1.4.1. ABC Transporters in Nutrient Transport across the BBB

Endothelial cells of the BBB are responsible for regulating the transport of essential nutrients, such as glucose, amino acids, and vitamins from the bloodstream into the brain [109,110]. This selective transport is essential for maintaining cerebral homeostasis and providing the brain with the necessary substrates for proper function. ABC transporters expressed in endothelial cells play a significant role in the process of facilitating the bidirectional movement of various nutrients across the BBB (Figure 1), such as the efflux of glutathione, an essential antioxidant, or glutathione-coupled metabolites by ABCC1 [91,111]. Moreover, ABCC1 and ABCC4 are also expressed in brain endothelial cells and contribute to nutrient transport across the BBB. ABCC4 on the other hand, is involved in the efflux of cyclic nucleotides and prostaglandins, which may have implications on brain signaling and metabolism [112]. Similarly, ABCB1, ABCG2, and ABCC1 are known to play a crucial role in maintaining the integrity of the BBB by restricting the permeation of neurotoxic chemicals. They allow the passage of necessary metabolic substrates and limit brain uptake of nutrients, ensuring proper brain function. Their selective transport mechanism is critical for nutrient delivery, as well as for preventing the accumulation of potentially neurotoxic substances in the brain [113].

#### 1.4.2. ABC Transporters in Neurovascular Coupling

Neurovascular coupling is the process by which changes in neuronal activity are coupled with changes in cerebral blood flow, ensuring that the brain receives an adequate blood supply to meet its metabolic demands [114]. Both endothelial cells and pericytes contribute to neurovascular coupling through the regulation of vascular tone and blood flow [115]. While the specific details on ABC transporters in neurovascular coupling have not been elucidated, it is understood that these transporters can influence this coupling indirectly through their control over the brain microenvironment. For example, ABC transporters expressed in endothelial cells modulate the production and release of vasoactive substances [109]. Specifically, ABCB1 and ABCG2 transporters have been shown to influence the release of endothelium-derived relaxing factors, such as nitric oxide, which regulates vascular tone [116]. Additionally, pericytes, which are closely associated with endothelial cells, express various ABC transporters involved in the regulation of pericyte contractility and vascular tone [117]. Moreover, indirect evidence for ABC transporter involvement in neurovascular coupling exists, such as for ABC transporters in AD pathogenesis, which is characterized by altered BBB permeability and neurovascular dysfunction. This alteration in transporter expression and function can contribute to the vascular Aβ pathology of AD by affecting nutrient transport and waste removal across the BBB [118]. It appears highly plausible that any dysfunction of these transporters could lead to altered neurovascular coupling and contribute to various neurological disorders [22].

In conclusion, ABC transporters expressed in endothelial cells and pericytes play critical roles in BBB integrity, nutrient transport across the BBB, and neurovascular coupling [63,119,120]. By regulating the efflux of toxic compounds, nutrient transport, and vascular tone, these transporters actively contribute to the proper functioning of the brain’s vasculature and maintenance of brain homeostasis. Dysregulation or dysfunction of endothelial and pericyte ABC transporters can have significant implications for BBB integrity, nutrient supply to the brain, and cerebral blood flow regulation in general, which may contribute to the pathogenesis of several neurological disorders [121]. Further research is needed to fully elucidate the specific mechanisms and therapeutic potential of these transporters in endothelial cells and pericytes.

**Figure 1 cells-13-00740-f001:**
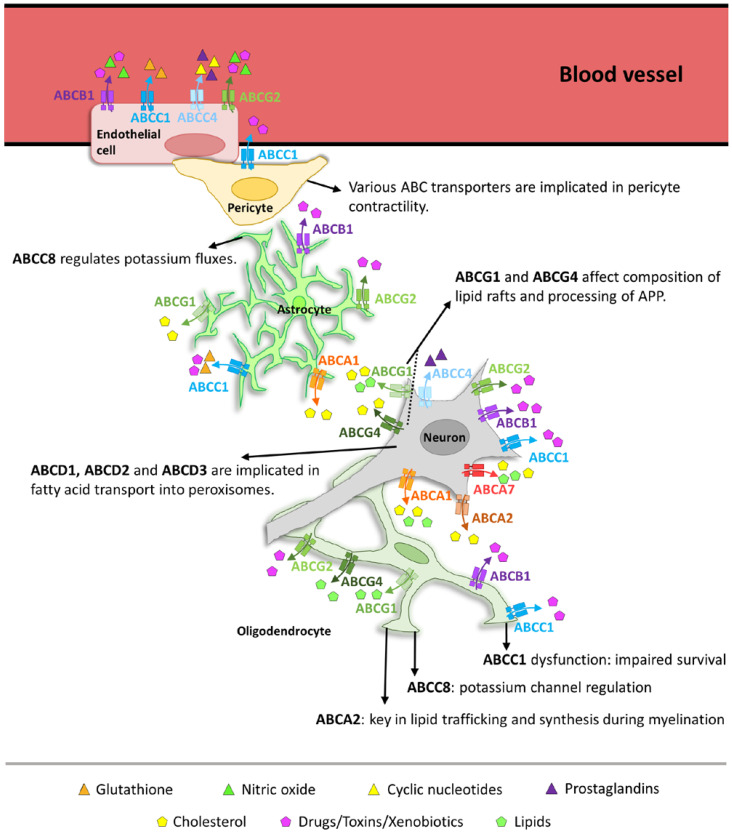
Graphical overview of the role of ABC transporters in different brain cells, such as glia, neurons, and vascular cells (endothelia and pericytes). The different molecules fluxed out of the cells are represented by different symbols, which are explained in the legend (at the bottom of the figure). The transporters implicated in the efflux of these substances are represented in different colors and specified for each cell. The black arrows originate at the cells in which the effects take place and defined actions or effects of certain transporters are specified at the arrowheads.

### 1.5. ABC Transporter Function in Immune Cells of the Brain

It has been widely accepted that the disruption of the BBB is a common feature in different neurological disorders, such as Multiple sclerosis (MS), AD, and PD [122,123,124], and leads to the infiltration of immune cells into the brain [125]. ABC transporters are differentially expressed on and have varying important functions in the various cells of the brain’s immune system (Figure 2), modulating the immune response and neuroinflammation caused by the invasion of peripheral immune cells and resident microglia.

#### 1.5.1. Lymphocytes/Monocytes

ABC transporters have been described to be involved in immune responses in several ways (Figure 2). They play a role in regulating lymphocytes and monocytes [126,127]; are implicated in modulating the immune response through cell maturation [128], activation [129], adhesion, and migration [126]; and perform their functions through the release of mediators, such as C-C motif chemokine ligand 2 (CCL2), platelet-activating factor (PAF), or prostaglandin E2 [127,130,131], or by general mechanisms such as cholesterol regulation [132,133,134,135]. The specific roles of the different ABC transporters in the immune response are further described:Role of ABC Transporters Related to T-cell FunctionsABCB1 is implicated in modulating the T-cell response, including cell infiltration, proliferation, and cytokine secretion. ABCB1 exerts these functions by interfering with dendritic cell function and maturation through the release of CCL2 [128]. Furthermore, ABCB1 plays a part in CD4^+^ and CD8^+^ T-cell migration and adhesion [126]. In addition, ABCB1 has an effect on CD8^+^ T-cell activation, expression, and generation of memory cell pools [129,136]. Inflamed endothelial cells, in which we find extensive ABCB1 expression, also secrete CCL2, affecting mainly CD8^+^ T-cell adhesion [126]. ABCG1 impacts inflammation mainly by regulating cholesterol. In the absence of ABCG1, cholesterol accumulates and, thereby, improves CD4^+^ T-cell receptor signaling. Furthermore, ABCG1 deficiency was shown to increase the absolute number of CD4^+^ T-cells, likely by the amplification of extracellular signal–regulated kinase (ERK) signaling. ABCG1 deficiency also enhances Treg (regulatory T)-cell development through their cholesterol content [135,136]. Inversely, ABCG1 upregulation is related to reduced T-cell proliferation [132,133,134,136].

Role of ABC Transporters Related to NKT-cell FunctionsABCA7 and ABCG1 are implicated in influencing NKT (natural killer T)-cell functions in immune responses. ABCA7 deficiency impedes the binding of NKT-cells and antigen-presenting cells (APCs), by CD1d accumulation in the late endosomal compartment, thereby altering NKT-cell development and activation [132,136]. In contrast to previous statements, there is evidence supporting the involvement of ABCG1 in the correct development of NKT-cells, as ABCG1 deficiency in NKT-cells resulted in a reduced number of these cells [132,133,134,136].

Role of ABC Transporters in Monocyte MigrationMonocyte migration is influenced by the ABCB1 transporter regulating the CCL2 release from reactive astrocytes [127]. Similarly, the ABCC1 transporter controls monocyte migration across the BBB due to its mediation of CCL2 release by reactive astrocytes [22,127].

Role of ABC Transporters in the Release of Inflammatory MediatorsABCB1, ABCC1, ABCC4, and ABCC8 have been implicated in the regulation of inflammatory mediators. Specifically, ABCC1 and ABCC4 are able to transport inflammatory mediators such as prostaglandin E2 [22,131]. Additionally, ABCC1 is involved in the leukotriene C4 (LTC_4_) release in all cells producing this mediator [22,92,137]. ABCC8 is implicated in altering inflammatory responses: the silencing of *Abcc8* alters SUR1-TRPM4 function in astrocytes, causing a reduced inflammatory response in EAE with a reduction in inflammatory infiltrates and inflammatory cytokines [138]. ABCB1 mediates the secretion of PAF by reactive astrocytes, thereby contributing to the modulation of the immune response [130].

**Figure 2 cells-13-00740-f002:**
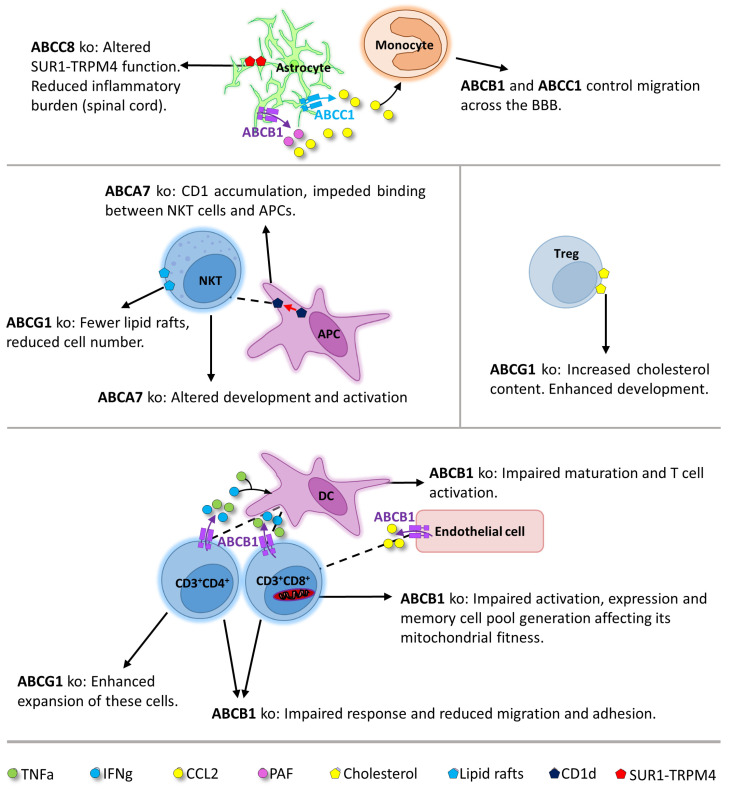
Graphical overview of the role of ABC transporters in maintaining brain immune functions by regulating lymphocyte, monocyte, and dendritic cell functions, also in collaboration with astrocytes. The different molecules fluxed out of the cells are represented by different symbols, which are explained in the legend (at the bottom of the figure). The ABC transporters implicated in the efflux of these substances are represented in different colors and specified for each cell. The black arrows originate at the cells in which the effects take place, and defined actions or effects of certain transporters are specified at the arrowheads. Discontinued lines represent interaction between cells.

#### 1.5.2. Microglia

ABC transporter expression has been described in microglia and macrophages in which they play essential roles in facilitating immune responses (Figure 3). Their function in microglia ranges from specifically modulating membrane cholesterol efflux [95,139,140,141] to the general efflux of diverse substrates [142,143]. The depletion of ABCA1 and ABCG1 in microglia has been shown to cause allodynia through the disruption of the formation of cholesterol-enriched rafts to which TLR4 co-localizes [144].

ABC Transporters in Microglia Cholesterol EffluxABCA1 expression, at both the mRNA and protein level, has been found in microglia. In ABCA1-deficient mice, a consequent decrease in cholesterol efflux from microglia is seen, including reduced APOE lipidation, thereby increasing the risk of AD [82,95,139,140]. In addition, an upregulation in *ABCG4* mRNA and ABCG4 protein levels in microglia located closely to Aβ plaques in human brains has been detected. As this transporter mediates cholesterol efflux, its upregulation improves APOE lipidation and could be considered a microglial protective response [82,141].

ABC Transporters in Aβ ClearanceABCA7 is highly expressed in microglia in which it is thought to play a role in modulating Aβ clearance. Deficiency of this transporter resulted in a reduced clearance of Aβ peptides [145,146].

ABC Transporters in General Substance Efflux and Drug ResistanceABCB1 expression was found in cell cultures of rat microglia, where it was localized at the plasma and nuclear membranes [82,147]. Its most probable role is thought to be the mediation of xenobiotics and toxic or general substances efflux [142,143]. This function has, for example, been documented by demonstrating that ABCB1 prevented the entry of digoxin in rat microglia cell cultures [142].In other studies, *Abcc1* mRNA and ABCC1 protein expression and function have been found in rat microglia cultures and the MLS-9 cell line [143,148,149]. Here, ABCC1 transporters played a role in the resistance to drugs, such as the chemotherapeutic vincristine [148]. The ABCC1 transporter is localized mainly in the plasma membrane, but also in caveolae and clathrin-coated vesicles [82,147,150].*Abcc4* and *Abcc5* mRNA, as well as ABCC4 and ABCC5 proteins, have been detected in rat microglia cultures [82,143,147,149,151]. Functional effects of both transporters in rat and mouse microglia cell lines have been implicated in the transport of antiretroviral 9-(2-phosphonylmethoxyethyl)adenine which is a substance used against the human immunodeficiency virus (HIV) [147,151].

ABCD1, a Transporter Directly Related to a DiseaseThe *ABCD1* gene codes for the ABCD1 protein, also known as the adrenoleukodystrophy (ALD) protein for its role in this specific disease [152]. Mutations affecting the ABCD1 transporter lead to ALD due to the dysfunctional transport of very-long-chain fatty acids into peroxisomes [82,152]. *Abcd1* mRNA and ABCD1 protein have been found in rodent CNS, and ABCD1 protein expression has been measured post mortem in human brains [82,152,153]. Specifically, ABCD1 is highly expressed in microglial cells [152]. Therefore, these cells could play a prominent role in the pathogenesis of the disease.

Other ABC Transporters*Abcc3* mRNA is expressed in rat microglia [82,147,149]. Nevertheless, no information about its functionality in these cells is available.*Abcc6* mRNA expression has been reported in newborn rat glial cell (including, but not exclusive to, microglia) cultures [82,154]. However, in another study, no *Abcc6* expression was found in adult rat cell cultures of microglia [149]. Therefore, a study focusing specifically on microglia cultures from newborn rats could help to elucidate if *Abcc6* mRNA is indeed present on and/or in these cells.ABCG2 protein expression has been found in rat microglia primary cultures. *Abcg2* mRNA and ABCG2 protein expression were also seen in a mouse microglia cell line (BV-2 cells), in which ABCG2 expression and function are downregulated by LPS-induced inflammation [82,147,155,156]. A reduction in ABCG2 function could lead to the accumulation of toxic substances in the brain [156]. The specific function of this transporter, however, was not demonstrated, probably due to the low expression in the BV2 cells or a lack of its localization to the plasma membrane [155].

In summary, the functions that various ABC transporters have in microglia have been mainly ascribed to their role in drug efflux and not to a direct impact on immune function.

#### 1.5.3. Macrophages

In macrophages, different ABC transporters have been described. As in microglia, these transporters can modulate cholesterol efflux [157,158,159,160,161,162,163,164], but are also implicated in phagocytosis [165] and Aβ clearance [145,166] (Figure 3).

Cholesterol Efflux and Lipid HomeostasisABCA1, ABCA5, ABCA6, ABCA9, ABCB4, ABCG1, and ABCG4 have been detected in macrophages, where they are involved in cholesterol efflux and lipid homeostasis [161,162,163,164,167,168,169,170]. It has been described and widely accepted that dysfunction of these transporters leads to an exacerbated immune response due to the dysregulation of these cells [157,159,160,169,170,171,172].

Phagocytosis and Aβ ClearanceABCA7 has been implicated in the phagocytosis of apoptotic cells by macrophages. The presence of this transporter on the cell surface of macrophages where it co-localizes with low-density lipoprotein receptor-related protein 1 (LRP1) is essential for ERK signaling [165]. Additionally, as stated for microglia, this transporter has also been proposed to play a direct role in Aβ clearance in macrophages [145,166].

**Figure 3 cells-13-00740-f003:**
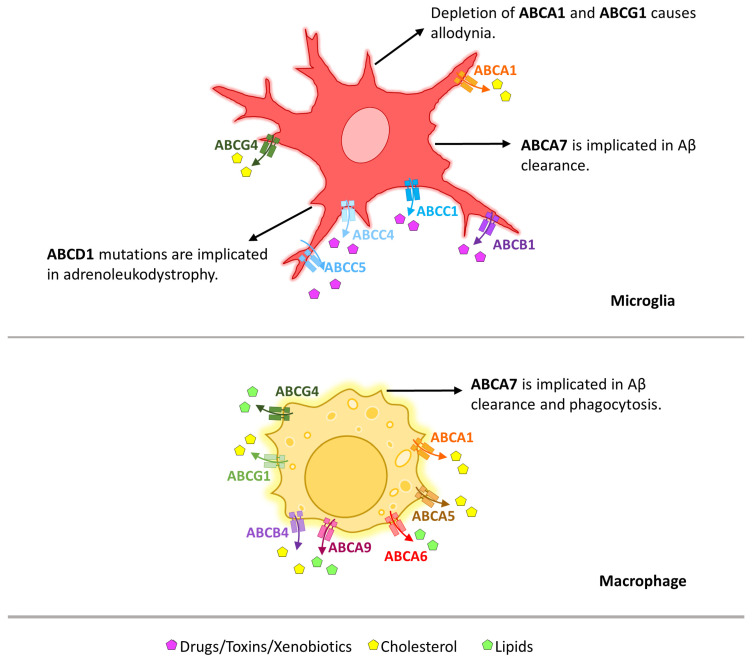
Graphical overview of the role of ABC transporters in microglia and macrophages. The different molecules fluxed out of the cells are represented by different symbols which are explained in the legend (at the bottom of the figure). The transporters implicated in the efflux of these substances are represented in different colors and specified for each cell. The black arrows originate at the cells in which the effects take place and defined actions of effects of certain transporters are specified at the arrowheads.

## 2. Function of ABC Transporters in Brain Diseases

### 2.1. ABC Transporter Dysfunction in Neurodegenerative Disorders

The important physiological role of ABC transporters for the correct functioning of the brain is undeniable. ABC transporters have a crucial function for transporting various substrates, as well as removing toxins and harmful molecules. Functional dysregulation of these transporters is associated with various diseases in several functional systems of our body, such as neurodegenerative, cardio-metabolic, and liver disorders [173,174,175,176,177,178,179]. Among these, neurodegenerative diseases are becoming an increasingly important and studied category. Counteracting the dysregulation of specific ABC transporters is emerging as a new therapeutic approach for some neurodegenerative diseases, such as AD, PD, and even extremely rare diseases such as frontotemporal dementia (FTD) in which ABCA2, ABCA3, ABCA4, ABCA7, ABCA9, ABCA10, and ABCA13 expression was found to be altered in patients with FTD with TDP-43 [180].

#### 2.1.1. ABC Transporters as Risk Factors for Alzheimer’s Disease

In AD, the accumulation of Aβ peptides in the brain is a key pathological hallmark of the disease. Cholesterol plays a role in Aβ metabolism. High levels of cholesterol were related to Aβ production, whereas cholesterol depletion was related to decreased Aβ levels [181,182]. ABC transporters are implicated in facilitating Aβ aggregation in different ways, specifically through:their participation in the lipid metabolism [183,184,185];their involvement in Aβ phagocytosis;Aβ trafficking into lysosomes [145,166,186].

ABC transporters such as ABCA1, ABCA7, ABCB1, ABCC1, ABCG1, and ABCG2 have been shown to play a critical role in the clearance of Aβ from the AD brain [27,28,187]. Mutations related to ABCA1 have been associated with AD. However, ABCA1 has also been implicated in AD resistance [188]. Furthermore, it has been described that ABCA1 overexpression or stimulation leads to a better prognosis in mice with AD [183,185,189]. ABCA2 has been implicated in AD pathogenesis, since its upregulation is related to an increase in APP synthesis [190]. In addition, ABCA2 depletion is associated with decreased Aβ production [184]. ABCA5 could play a protective role in AD as an increase in its expression is related to a decreased Aβ load in vitro [189,191].

As mentioned above, ABCA7 participates in Aβ clearance and production [56,145,166,175,187]. Coding sequence mutations and promoter and intron nucleotide polymorphisms related to the human *ABCA7* locus have been associated with an increased risk for AD, promoting and highlighting ABCA7 as a new significant risk marker besides APOE4 [188,189,192,193,194,195].

ABCB1, ABCC1, and ABCG2 also play a role in AD progression and Aβ clearance. Therefore, targeting these transporters could be a potential therapeutic strategy [196,197].

In general, the dysfunction of specific ABC transporters can lead to the accumulation of Aβ in the brain, which can contribute to the development and progression of AD. Modulating ABC transporters has been previously suggested as a possibility for AD therapeutics in mice [12,198]. However, it has to be strongly emphasized that export via the BBB or BCPB facilitated through ABC transporters, e.g., mouse Abcc1, requires additional transporters, such as Oat3 and Abcc4/Mrp4 [199]. The latter are expressed very differently between mice and human, some even fully lacking expression in humans. These important species differences will play a confounding and possibly also detrimental role, hampering the development of drugs and PET tracers using ABC transporters.

ABCA1-APOE AxisAn impairment of the ABCA1- and SR-BI-mediated cholesterol efflux pathways and HDL anti-inflammatory activity has already been reported in AD [200].Different *ABCA1* genetic variants related to the risk of developing AD have been described. For example, the SNPs rs2230806 and rs2230808 have been associated with the risk of AD in Caribbean Hispanics using APOE-conditional models [201,202]. Furthermore, the SNPs rs2230805 and rs2230806 have been characterized as important markers for different forms of dementia, including AD, in a Swedish cohort [203]. Moreover, some polymorphisms in the coding region (R219K, I883M, and R1587K) or in the promoter region (C-14T) of *ABCA1* were related to a higher risk of developing AD in a Spanish cohort [204]. In this cohort, it was also found that the *ABCA1-14T* allele increased the risk of AD synergistically when acting together with the *APOE4* allele, probably by increasingly inefficient cholesterol efflux from glial cells [204]. Furthermore, an association between a rare loss-of-function variant in ABCA1 (N1800H) and increased risk of AD was found [205]. Some microRNAs which regulate *ABCA1* have been implicated in Aβ accumulation (miR-106b) [206] or cholesterol metabolism (miR-758 and miR-33) [207,208,209,210].Modulating ABCA1 function showed its functional impact in AD models. It has been described that the ABCA1 agonist mifepriston improved aggregation pathologies driven by APOE4 [211]. Earlier findings by Koldamova et al. revealed that the absence of ABCA1 significantly reduces the levels of APOE in the brain [212]. With lower levels of APOE, there is a notable increase in Aβ deposition in the brains of APP23 mice. This suggests that ABCA1 is essential for maintaining normal APOE levels in the brain, which, in turn, affects the clearance of Aβ [212]. Wahrle et al. investigated the potential therapeutic role of targeting ABCA1 in AD [183]. Their study, based on the PDAPP mouse model, found that enhancing the function of ABCA1 through overexpression (*mPrP-mAbca1*) led to a significant reduction in Aβ accumulation in the brain. The study also demonstrated that enhancing ABCA1 function led to the increased lipidation of APOE-containing vesicles and clearance of Aβ from the brain [183].

ABCA7There are several ABCA7 gene variants associated with an increased risk for AD. Most of these variations influence the risk for AD depending on ethnic background. In general, ABCA7 loss-of-function variants lead to increased AD risk [192,193,213,214,215]. A putative protective coding variant (p. G215S) was identified in people of British and North American ancestry [216]. Another genomic study found that increased ABCA7 expression due to the rs3764650T allele reduces the risk of AD, suggesting that dysfunction of the ABCA7 transporter may contribute to the accumulation of Aβ in the brain [217].

ABCB1It has been demonstrated that Aβ is transported by ABCB1 in vitro [218], and Aβ clearance deficiency is associated with a low expression of the transporter in vivo [11,29]. Thus, the ABCB1 transporter seems to be directly implicated in AD pathogenesis. At the same time, Aβ reduces the expression of ABCB1, leading to increased Aβ deposition and exacerbated disease [11,219]. A study investigating the potential therapeutic role of targeting ABCB1 ubiquitination in Tg2576 mice demonstrated that preventing ubiquitination enhanced ABCB1 function and led to the increased clearance of Aβ, resulting in decreased Aβ brain levels [220]. Another study, a post mortem neuropathological study suggested that dysfunction or the decreased expression of ABCB1 could lead to the build-up of Aβ in the walls of cerebral blood vessels, contributing to the pathogenesis of CAA [11]. In summary, many findings to date emphasize the role of ABCB1 in regulating Aβ levels in the brain and, thereby, its potential relevance for understanding the early mechanisms of AD development.

ABCC1ABCC1 has been implicated as an important early modulator of AD onset, already before the deposition of Aβ starts. Krohn et al. [197] studied different mouse models lacking various ABC transporters (ABCB1 or ABCC1) and/or the protease neprilysin (NEP), a major Aβ-degrading enzyme. NEP deficiency itself induces higher endogenous mouse Aβ levels. Mice lacking both ABCC1 and NEP showed a marked early astrocyte reaction upon increased mouse Aβ, specifically in the dentate gyrus and amygdala regions, the locations where AD pathology is first seen in patients. Thus, this NEP × ABCC1 double-knockout model represents the first non-transgenic, preclinical AD mouse model [197].It has also been described that ABCC1 deficiency leads to increased Aβ levels in APPtg mice (APPPS1-21 model), while its activation, using thiethylperazine, reduces Aβ load [12]. These results suggest that ABCC1 transporters may play a role during AD development and could be a target for the modulation of disease onset, as well as a diagnostic tool for functional PET imaging [221,222].

ABCG2ABCG2 was shown to directly interact with Aβ in AD. For example, a study revealed that this transporter controls the entry of peripheral Aβ into the CNS [196]. It was shown that ABCG2 is upregulated in human cerebral vessels and in the AD mouse brain, and that its dysfunction may contribute to the accumulation of TAU in the brain [223]. The ABCG2 C421A polymorphism has been associated with an increased susceptibility to AD, particularly when combined with the APOE4 allele [224]. ABCG2 has been suggested to reduce toxicity and inflammation in the brain, which are key components in the pathology of AD [225]. Moreover, studies have found that the brain distribution of dual ABCB1/ABCG2 substrates is unchanged in an AD mouse model (APPPS1-21), indicating that the brain distribution of clinically used ABCB1/ABCG2 substrate drugs may not differ between AD patients and healthy individuals [226].

In summary, recent research has highlighted the importance of ABC transporters in the clearance of Aβ and possibly also TAU, two proteins that are implicated in the pathogenesis of AD [227,228]. Targeting these transporters may offer a promising therapeutic strategy for the treatment of AD. To assess potential therapeutic implications of ABC transporters, further research is needed. This will also lead to a better understanding of the precise roles of these transporters in the clearance of Aβ and TAU.

#### 2.1.2. ABC Transporters in Movement Disorders

α-SynucleinopathiesABCB1 plays a role in α-synucleinopathies, such as PD. Mutations related to *ABCB1* have been associated with the risk of developing PD. For example, in an ethnic Chinese population, genetic ABCB1 variations [SNPs—e12/1236(C/T), e21/2677(G/T/A), and e26/3435(C/T)] were shown to increase the risk of PD [229]. However, this genotype did not reach statistical significance in European populations [230,231]. In another study, also in an ethnic Chinese population, genetic variation regarding *ABCB1* (SNPs e21/2677T and e26/3435T) was associated with protection against the risk of developing PD [232].On the other hand, exposition to pesticides is a risk factor for developing this pathology. This risk becomes especially noticeable in people carrying the 3435T allele of the *ABCB1* gene, which is associated with a decreased pump function [233].Using PET (^11^C-Verapamil tracer), Bartels and colleagues demonstrated the decreased BBB function of ABCB1 in disease-specific brain regions at later disease stages of PD, as well as multiple system atrophy (MSA) and Lewy body dementia (LBD) [19,234]. It was proposed that the accumulation of toxic substances, such as pesticides but also aggregated proteins, due to the loss of ABCB1 function associated with aging, is a key incident in the development of PD and other neurodegenerative diseases [18,235].

Huntington’s DiseaseStudies have indicated that cholesterol homeostasis may be disrupted in individuals with HD, potentially influencing disease progression. These abnormalities in cholesterol metabolism could impact neuronal function and, thus, contribute to the neurodegenerative processes seen in HD [20,236]. Given the abundant amount of evidence delineating the regulation of cholesterol by ABC transporters [20,237,238,239], the respective transporters involved in cholesterol efflux may be targets for HD modulation and treatment. For example, silencing astrocytic ABCA1 resulted in poor neurite outgrowth in HD neurons [21]. Furthermore, ABCA1 and ABCG4 are reduced in primary astrocytes from R6/2 and YAC128 mice, two mouse models mimicking many features of HD [20]. A study using the R6/2 mouse model found that ABCB10 is also implicated in modulating HD pathology, participating in the UPRmt pathway. ABCB10 is downregulated in HD cells, and its depletion is related to cell death and the production of reactive oxygen species [173]. On the other hand, ABCB10 overexpression has shown the opposite effect, presenting itself as a therapeutic opportunity [173]. Moreover, ABCD1 might be involved in HD due to its role in reacting to oxidative stress, which is a key factor in HD pathology [240].A transcriptome-wide association study (TWAS) identified several genomic modifiers of HD onset [241]. The findings of this study indirectly highlight the significance of exploring ABCA7 in HD, given its prominent association to AD [189] and its close genetic location 5′ of the detected *HMHA1* locus. This connection implies the general potential relevance of ABCA7 for the modulation of neurodegenerative mechanisms. At the same time, the genomic region encompassing *HMHA1* and *ABCA7* emerges as a critical area for further research into the molecular underpinnings of HD [241].The exact mechanisms and specific roles of ABC transporters in HD are still under investigation by us. Utilizing ABC transporter ko mice to examine modulatory effects of these transporters addresses a critical knowledge gap in the understanding of the role of ABC transporters in HD onset and progression. Very recently, the longitudinal characterization of ABCA7-deficient HD mice showed delaying effects on HD onset and progression in a sex-dependent manner. An understanding of the molecular mechanisms could have implications for new therapeutic strategies. Modulating the ABC transporter activity may offer novel approaches for addressing abnormal or dysfunctional lipid metabolism, protein transport, and neurodegenerative processes associated with HD. While some studies suggest altered cholesterol levels or metabolism in HD patients [242], the mechanisms involved are still the subject of investigation, especially to differentiate whether these changes are a cause or consequence of the disease. Further research is needed to uncover the precise mechanisms and therapeutic potential of targeting ABC transporters in HD.

### 2.2. Other, Non-Inflammatory Brain Diseases

#### 2.2.1. White Matter Disorders—Adrenoleukodystrophy ABCD1

As stated before, mutations in *ABCD1* are directly implicated in X-linked adrenoleukodystrophy. Mutations affecting *ABCD1* cause an accumulation of very long-chain fatty acids in the brain, the spinal cord, and the adrenal gland cortex, leading to neurological and endocrine dysfunctions. More than 800 mutations have been described for this disease [243,244,245].

#### 2.2.2. ABC Transporters in Epilepsy

The atypical expression and/or function of ABC transporters are assumed to contribute to refractory epilepsy [246], likely due to the role of inflammation on ABC transporter modulation [247,248]. Furthermore, the expression and activity of efflux transporters are increased, both in epilepsy patients and animal models of epilepsy, thereby counteracting the increased CNS permeability typical of this pathology [23,249,250,251,252,253,254]. This increase plays an important role for epilepsy treatment strategies. Since these transporters are implicated in drug efflux, anti-epileptic treatment becomes less effective [249,255,256,257,258,259,260,261]. Additionally, ABC transporters may also play a direct role in the development of pathologies related to epilepsy, such as hippocampal sclerosis (HS) and cortical dysplasia (CD) [24,251,262].

##### Hippocampal Sclerosis of Epilepsy and Aging

Hippocampal sclerosis is a common pathology encountered in mesial temporal lobe epilepsy and other epilepsy syndromes, and is defined by neuronal loss and reactive astrocyte proliferation [263]. In drug-resistant epilepsy patients, the overexpression of ABCB1 and ABCC1 was found on reactive astrocytes and the overexpression of ABCB1-3 and ABCC4 was detected on endothelial cells [249,264,265,266].

Hippocampal sclerosis of aging (HS-aging) is related to a genetic risk locus, which is an *ABCC9* SNP pair (rs704178 and rs704180). Loss of function of the ABCC9 transporter, which is located on pericyte membranes, contributes to the disruption of the neurovascular unit followed by the development of cognitive impairment [267,268].

Cortical DysplasiaCortical dysplasia [269] arises from abnormal stem/progenitor cell function, in which ABC transporters are involved [270,271], and could account for seizure relapse after epilepsy surgery [272]. ABCB1 and ABCC1 have been detected in human tissue samples of focal cortical dysplasia and benign tumor lesions in therapy-refractory epilepsy patients [23,262,273].

Resistance to TreatmentABC transporters are considered the most clinically important drug efflux pumps in drug-resistant epilepsy [249]. ABCB1 plays a role in oxcarbazepine resistance, as evidenced by the finding that brain levels of this transporter in treated epilepsy patients are inversely proportional to the levels of oxcarbazepine active metabolites [71,274]. Further corroborating this relationship, it has been shown that the co-administration of an ABCB1 inhibitor together with oxcarbazepine leads to a decrease in pilocarpine-induced limbic motor seizures and seizure severity [261]. In animal models, ABCB1 has also been proven to mediate the access of other antiepileptic drugs to the brain [275,276,277,278,279]. In humans, *ABCB1* has been shown to be overexpressed in brain tissue from medically intractable epilepsy patients, and the transporter is assumed to mediate the export of the antiepileptic drug phenytoin from the brain, leading to inadequate antiepileptic drug levels [280]. ABCB1, but also ABCC1, ABCC2, and ABCC5, expression is increased in brains from medically intractable epilepsy patients, hinting at their role in the resistance to treatments [23,24,67,249,262,280]. In fact, the co-administration of antiepileptic treatments and ABCC1 inhibitors has been shown to improve the efficacy of antiepileptic drugs such as oxcarbazepine [261], as well as increasing the levels of phenytoin [281] and carbamazepine [276]. The same has been evidenced regarding ABCC2 inhibition, which significantly improved the anticonvulsant effect of the antiepileptic drug phenytoin [282]. Of note, the excessive use of some antiepileptic drugs may induce an increased expression of ABC transporters, actually contributing to the development of drug-resistance [249,283,284,285].

### 2.3. ABC Transporter Function in Neuroinflammatory Diseases

#### 2.3.1. Multiple Sclerosis

MS is a chronic autoimmune neuroinflammatory disorder characterized mainly by CNS white matter damage and secondary neurodegeneration affecting the grey matter, as reviewed in [286]. It is the most common chronic inflammatory demyelinating disease of the CNS [287,288] and one of the most common causes of neurological disabilities in young adults [289]. MS is considered an autoimmune disease in which symptoms are caused by the damage to whiter matter tracts and, thus, are related to the affected regions. The symptoms have been widely reviewed and usually include sensory, motor, and visual deficits, but also include autonomic problems such as bladder dysfunction and cognitive and affective disorders [290,291,292]. MS can be categorized into different forms, largely depending on the onset and evolution of symptoms: relapsing–remitting (RRMS), primary progressive (PPMS), or secondary progressive (SPMS) [288,290,293]. Its pathology is mainly caused by inflammation and the resulting demyelination. MS is characterized by an inflammatory process triggering a cascade of further pathological events. Briefly, activated T-cells infiltrate into the CNS and differentiate, activating and attracting more inflammatory cells. B-cells, monocytes, and macrophages also migrate through the BBB, participating in the inflammatory process. This inflammation leads to demyelination by directly damaging oligodendrocytes, which, then, is followed by secondary axonal loss. During this process, accompanying astro- and microgliosis occurs, thus astrocytes and microglia are involved in this pathology [288,294,295]. It is thought that relapses are triggered by or are related to inflammatory events which cause inflammatory cell infiltration waves into the CNS. Long-term disability due to MS is caused by the ultimate irreversible axonal damage and neuronal loss [288,294].

##### Alterations of ABC Transporters Modulating MS

To study the role of ABC transporters and their alterations in the progression and treatment of MS, researchers have used tissue from patients [126,127,296,297]. The course of human MS can be recapitulated to some extent in a rodent model of the disease *experimental autoimmune encephalomyelitis* (EAE). Reproducing this pathology in rodents appears to be a good and replicable option [126,128,296,298]. In this regard, it is important to emphasize the diverse functions that ABC transporters can facilitate from modulating cholesterol efflux, acting as drug transporters and participating in immune response modulation (see Section 1.5). The roles that different ABC transporters might play in MS are further described below.

Stearoyl-CoA desaturase-1 (SCD1) is an essential enzyme for the conversion of saturated into mono-unsaturated fatty acids. In MS, the sustained accumulation of myelin in phagocytes promotes a shift in their reparative phenotype through SCD1. It has been reported that an increase in SCD1 leads to a reduction of ABCA1 cell surface expression, affecting cholesterol efflux and promoting lipid accumulation, thus resulting in an inflammatory phagocyte phenotype [299]. In addition, SCD1 inhibition or deficiency prevented ABCA1 reduction and correlated with an improvement in re-myelination [299].

Regarding ABCB1, the drug-induced functional downregulation in endothelial cells (at the BBB) in active and inactive lesions of EAE revealed beneficial effects on nerve loss [300]. The loss of ABCB1, as seen in astrocytes, might play a beneficial role for treatment, improving CNS access for some medications. On the other hand, this reduction may also be implicated in increased neurotoxicity [127].

ABCC1 is increased in foamy macrophages and reactive astrocytes, where it plays a role for their CCL2-secretion during the course of MS, which has a detrimental role [127]. Targeting astrocytic SUR1-TRPM4/ABCC8 could be a promising therapeutic approach due to its inhibition during EAE [138,301].

On the other hand, it has been described that peroxisomal membrane protein 70 (ABCD3, PMP70) is reduced in the grey matter of MS patients. This alteration could be related to direct peroxisomal changes affecting MS onset and development [297].

Notably, RRMS patients have a diminished cholesterol efflux via ABCG1, both at serum and cellular levels. Furthermore, these patients also present with decreased monocyte mRNA levels of *ABCG1*. A reduction in cholesterol efflux due to the dysregulation of ABCG1 might result in a disproportionate immune response and an exacerbated progression of the disease [302] since altered cholesterol homeostasis is implicated in inflammation and cellular inflammatory phenotypes.

On the other hand, ABCG2 expression in foamy macrophages in MS was found to be increased [127]. This transporter plays a role in the teriflunomide treatment of EAE in mice. Here, it has been shown that, in ABCG2-deficient mice, therapeutic effects of the treatment were observed after using a lower dose of the drug compared to mice with a functional ABCG2 transporter [303]. Furthermore, the ABCG2 transporter is also implicated in mitoxantrone treatment, which is used as an escalation therapy in highly active MS [304]. Therefore, the modulation of ABCG2 could help to improve the treatment effectiveness.

In summary, ABC transporters appear to be crucially involved in various aspects of MS pathology, and, thus, their regulation and modulation could present a promising strategy to better study and understand, as well as more effectively treat, this disease.

#### 2.3.2. Detecting Clinical Effects of ABC Transporter Deficiency in EAE, a Murine Model of MS

To examine the effects and roles of different ABC transporters in neuroinflammatory diseases, we employed the murine model for human MS, EAE. Active EAE was induced in ABCB1a/b, ABCC1, and ABCA7 knockout mice, respectively, using age- and gender-matched C57BL/6J mice as controls.

ABCB1a/b, ABCC1, and ABCA7 deficiency all lead to a significantly reduced disease burden and this improved the clinical EAE course in the respective ko lines as compared to C57BL/6 controls, starting in and lasting for all or much of the early effector phase (Figure 4, Figure 5 and Figure 6, [305]). Regarding ABCB1 deficiency, our results, thus, recapitulate and corroborate previous data by Kooji et al. who also demonstrated an improved clinical EAE course in the absence of the ABCB1 transporter [128].

In summary, our results confirm the involvement of these three ABC transporters in the modulation of induced neuroinflammation. However, the precise mechanisms underlying this modulation need to be elucidated. One can assume that these are related to the known roles of ABC transporters in the immune response as described in Section 1.5 of this article. Our results also point to ABCB1, ABCC1, and ABCA7 as potentially useful targets for modulating the MS disease course in patients.

## 3. General Physiological Effects of ABC Transporters on Modifying Diseases

### 3.1. Export of Toxic Metabolites/Peptides

ABC transporters play a crucial role as defensive elements by transporting toxic metabolites from the space surrounding the brain vessels, e.g., the glymphatic space, back into the blood flow. Their role in the clearance of toxic metabolites is important in a number of neurodegenerative disorders, such as AD and PD, and has been described in detail elsewhere [19,235,306,307,308].

Other less recognized findings include the following:The amount of the ABC transporter ABCB1 was found to be reduced in Creutzfeldt–Jakob Disease (CJD) and could be involved in removing misfolded prion protein [309].The development of alimentary Progressive Supranuclear Palsy (PSP) from *Annonaceae* fruit intake has direct neurotoxic effects, but also leads to the accumulation of harmful molecules through ABCB1 inhibition [310,311].

### 3.2. Examples of Molecular Mechanisms Influenced by ABC Transporters

ABC transporters are crucial in several cellular compartments, from plasma membrane, lysosomes, and peroxisomes, to mitochondria, where the disruption of their function can influence various molecular pathways/mechanisms and induce diseases.

As already discussed, dysfunction of ABCD1 leads to ALD due to a defective transport of very-long-chain fatty acids into peroxisomes, hindering their degradation [312].Mutations in *ABCD4* alter vitamin B12 metabolism due to a failure of its release from lysosomes [313].ABCB9 was implicated in the release of peptides from lysosomes [314].ABC B subfamily transporters are implicated in mitochondrial mechanisms:○ABCB6 is implicated in porphyrin transport into mitochondria and heme biosynthesis [315,316]. Its deficiency is associated with increased mortality in mice under extreme demand in porphyrin conditions [317].○The ABCB7 transporter is essential for iron metabolism. It is needed for hematopoiesis, and its mutation causes X-linked sideroblastic anemia and ataxia [318,319,320,321].○ABCB8 is also implicated in iron mechanisms, particularly in iron export from mitochondria [322].○ABCB10 is found in mitochondria handling ROS [323].ABC transporters of the ABC C subfamily have been linked to the functionality and modulation of ion channels. For instance, ABCC7, also known as cystic fibrosis transmembrane conductance regulator, is a chloride channel itself [324]. In addition, ABCC8 and ABCC9, also known as SUR1 and SUR2, respectively, are modulators of potassium channels [325,326,327].The role of several ABC transporters, mainly of the ABC A subfamily, is widely recognized for the lipid transport of various lipid classes [328,329]. Involvement in lipid transport provides an immediate influence on all pathways of the cell’s signaling systems through direct interaction with signaling molecules, e.g., fatty acids or steroid hormones, or through indirect interaction via the fluidity of membranes, e.g., cholesterol content and the movement/clustering of transmembrane receptors. ABCA1 has been described to transport cholesterol and phospholipids, essential membrane lipids [330,331]. ABCA2 has been suggested to play a role in sterol transport [332]. ABCA3 is implicated in lipid metabolism and phospholipid homeostasis for surfactants [333,334,335]. ABCA4 has been described to import N-retinylidene-phosphatidylethanolamine and phosphatidylethanolamine [43].

The link between general molecular processes involving lipids, especially membrane lipids, and sterol-related mechanisms is important to address and will be further described in the next section.

#### 3.2.1. Modulation of Steroid Hormone Signaling by ABC Transporters

Steroid hormone regulation has been widely reviewed to play a role in the development of neurodegenerative diseases [336,337]. Sex hormones control neuron proliferation and differentiation (anabolic effects), and changes in hormone balances during andropause and menopause are linked to neurodegenerative diseases [338,339,340]. For example, changes in estradiol levels during the female lifespan are thought to be related to the increased risk of AD in women as compared to men [341].

Some ABC transporters are able to modulate the traffic of steroid hormones and their metabolites [342,343]. ABCB1 has been shown to transport cortisol and aldosterone [344]. The ABC C subfamily is implicated in steroid hormone regulation. ABCC1 plays a role in regulating steroid hormone homeostasis, and the lack of ABCC1 causes a reduction in concentrations of testicular steroids [345]. Moreover, ABCC1 transports estrone 3-sulfate and dehydroepiandrosterone 3-sulphate (DHEAS) in the presence of glutathione [346,347]. ABCC3, but also ABCC2, has been shown to transport some metabolites of testosterone and androgen glucuronides [348,349]. For instance, ABCC3 transports estrogen, β-estradiol 17-(β-D-glucuronide) (E217βG), and dehydroepiandrosterone (DHEA), which is the precursor molecule of different sex hormones, including testosterone and estradiol [350,351]. It has been found that ABCC4 also transports DHEA and conjugated steroids [347]. Furthermore, ABCC8 also transports steroid sulfates such as DHEA and E217βG [352]. ABCG2, however, has been demonstrated to transport sulfate-conjugates of estrogens, but not free estrone nor 17β-estradiol [353].

It is important to consider the relevance of cholesterol in the metabolism of steroid hormones, since it can serve as a precursor for the synthesis of steroid hormones or neurosteroids [354]. The intriguing findings of disease-delaying effects in male and even disease-preventing effects in female HD mice (model zQ175dn [239]) upon the knockout of ABCA7 function demand a more intense exploration of the multifaceted roles of ABC transporters, particularly those belonging to the ABC A subfamily. Beyond their established functions in lipid transport and lipid homeostasis [42], emerging evidence suggests potential links between ABC transporters and the steroid hormone signaling modulation of the body’s anabolic vs. catabolic balance. Future investigations exploring the specific mechanisms by which ABC A subfamily transporters impact steroid hormone pathways may yield valuable information for developing targeted interventions in the context of HD and other illnesses.

## 4. Summary

In conclusion, we here reviewed the current knowledge about the expression and function of different ABC transporters in glial and various other cells in the CNS. We, furthermore, delineated the importance of ABC transporters for several neurodegenerative and also neuroinflammatory diseases. Finally, we illustrated the promising roles that modulating the expression and function of these transporters could have on the treatment and development of neurodegenerative and neuroinflammatory CNS diseases.

## Figures and Tables

**Figure 4 cells-13-00740-f004:**
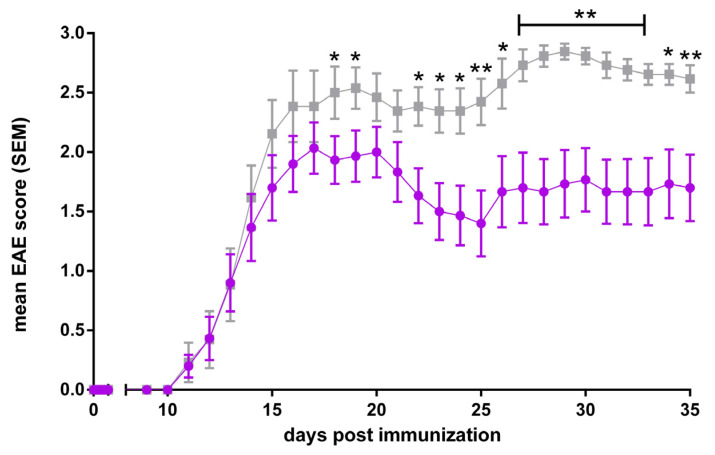
Clinical course of EAE in ABCB1a/b-ko mice (purple) compared to C57BL/6J control mice (grey). Data presented in the figure (mean +/− SEM) were statistically analyzed using a Mann–Whitney U test and Wilcoxon rank-sum test. * *p* < 0.05; ** *p* < 0.01. ABCB1-ko n = 15; C57BL/6J n = 13.

**Figure 5 cells-13-00740-f005:**
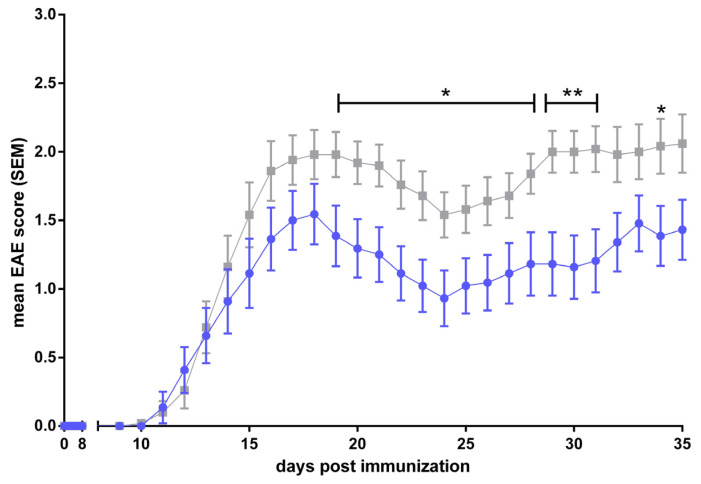
Clinical course of EAE in ABCC1-ko mice (blue) compared to C57BL/6J control mice (grey). Data presented in the figure (mean +/− SEM) were statistically analyzed using a Mann–Whitney U test and Wilcoxon rank-sum test. * *p* < 0.05; ** *p* < 0.01. ABCC1-ko n = 22; C57BL/6J n = 25.

**Figure 6 cells-13-00740-f006:**
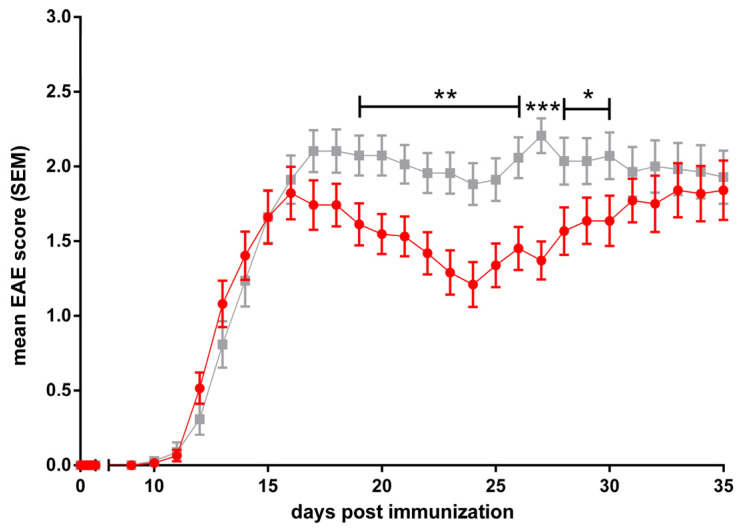
Clinical course of EAE in ABCA7-ko mice (red) compared to C57BL/6J control mice (grey). Data presented in the figure (mean +/− SEM) were statistically analyzed using a Mann–Whitney U test and Wilcoxon rank-sum test. * *p* < 0.05; ** *p* < 0.01; *** *p* < 0.001. ABCA7-ko n = 31 until day 27 and n = 22 from day 28; C57BL/6J n = 34 until day 27 and n = 28 from day 28.

## Data Availability

The data files and figures can be downloaded from the ‘Pahnke Lab-open access’ project at https://dx.doi.org/10.17605/OSF.IO/VWQ58.

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
