# Peer review of "Emerging Role of ABC Transporters in Glia Cells in Health and Diseases of the Central Nervous System"

_cells, 2024, doi:10.3390/cells13090740_

Round 1

Reviewer 1 Report

Comments and Suggestions for Authors

The manuscript by Villa et al. explores the role of ATP-binding cassette (ABC) transporters across various cellular membranes under physiological and pathological conditions. The authors provide a detailed analysis of the patterns of expression and function of ABC transporters, organizing the information in a manner beneficial to the reader. However, there are significant concerns regarding the grammar and fluidity of the text, which require thorough revision, including the elimination of repetitive sentences. Additionally, it would be constructive to add a paragraph elucidating the potential molecular mechanisms through which abnormalities in ABC transporters may contribute to the onset of neurodegenerative diseases

Comments on the Quality of English Language

there are significant concerns regarding the grammar and fluidity of the text, which require thorough revision, including the elimination of repetitive sentences.

Author Response

The manuscript by Villa et al. explores the role of ATP-binding cassette (ABC) transporters across various cellular membranes under physiological and pathological conditions. The authors provide a detailed analysis of the patterns of expression and function of ABC transporters, organizing the information in a manner beneficial to the reader.

However, there are significant concerns regarding the grammar and fluidity of the text, which require thorough revision, including the elimination of repetitive sentences.

The text was fully revised and corrected for language (English). We added more information and citations. Sentences were shortened and revised for better reading. Repetitive information was removed.

Additionally, it would be constructive to add a paragraph elucidating the potential molecular mechanisms through which abnormalities in ABC transporters may contribute to the onset of neurodegenerative diseases.

We added a separate section about examples of molecular interactions and regulation of ABC transporters in paragraph 3.

Comments on the Quality of English Language:

There are significant concerns regarding the grammar and fluidity of the text, which require thorough revision, including the elimination of repetitive sentences.

          The text was fully revised and corrected for English.

Reviewer 2 Report

Comments and Suggestions for Authors

This review paper well describes recent important topics concerning ATP-driven ABC transporters in glial cells, which are implicated in various diseases, including Alzheimer’s and Huntington’s diseases, and so on. It makes a significant contribution to the field and is well-organized and comprehensively described.

Minor points:

  1. 1. In Fig. 2, the expression "ABCG1 k.o.:" may not be easily understandable for all readers. It would be beneficial to include a brief explanation of "knock out (k.o.)" in the figure legend or the main text.
  2. 2. Additionally, in Fig. 2, the phrase "ABCG1 k.o: Enhanced expansion" may not be immediately clear to readers. Providing clearer labeling or an explanatory note in the legend would improve understanding.
  3. 3. In Figs. 1-3, the color code for the transported molecules is not easily distinguishable. For example, it is difficult to differentiate between PAF, cholesterol, lipid rafts, and cd1d. Enhancing the color difference or providing clearer distinctions would help readers.
  4. 4. Moreover, in Figs. 1-3, the figure legends are not sufficiently helpful for readers. Providing a more detailed figure legend or explaining the figures in the text should be essential for readers.

Author Response

This review paper well describes recent important topics concerning ATP-driven ABC transporters in glial cells, which are implicated in various diseases, including Alzheimer’s and Huntington’s diseases, and so on. It makes a significant contribution to the field and is well-organized and comprehensively described.

Minor points:

  1. In Fig. 2, the expression "ABCG1 k.o.:" may not be easily understandable for all readers. It would be beneficial to include a brief explanation of "knock out (k.o.)" in the figure legend or the main text.

In Fig 2 we changed "k.o." for "ko" and added that abbreviation to the main text and to the abbreviations list.

  1. Additionally, in Fig. 2, the phrase "ABCG1 k.o: Enhanced expansion" may not be immediately clear to readers. Providing clearer labeling or an explanatory note in the legend would improve understanding.

In Fig 2 I changed "Enhanced expansion" for "Enhanced expansion of these cells" to provide a clearer labelling.

  1. In Figs. 1-3, the color code for the transported molecules is not easily distinguishable. For example, it is difficult to differentiate between PAF, cholesterol, lipid rafts, and cd1d. Enhancing the color difference or providing clearer distinctions would help readers.

In Figs 1-3 I changed the color code of the molecules and made them bigger to make them more distinguishable.

  1. Moreover, in Figs. 1-3, the figure legends are not sufficiently helpful for readers. Providing a more detailed figure legend or explaining the figures in the text should be essential for readers.

In Figs 1-2 I added further explanation of the figures to the footnotes

The text was fully revised and corrected for English.

All figure legends were all adopted together with the text in the figures, see also above. We corrected for writing errors and revised terms to enable easier reading.

Reviewer 3 Report

Comments and Suggestions for Authors

The manuscript reviews the family of ABC transporters role and function in glial cells as well as other cells within the CNS system, focusing the attention on their role in several neurodegenerative disease. The work is  comprehensive, with many lights and few shadows: among the lights are the extensive citation and an effort to cluster relevant articles published by thematic/functionality of the ABCs transporters, shadows include its somehow superficial breadth since does not dig in depth on the molecular mechanisms proposed for their role in the original articles they review. Nevertheless I believe it constitutes a good starting point for all researches that begin to approach this rather complex field, to get handy and quick overview of the relevant literature of the field. 

Author Response

The manuscript reviews the family of ABC transporters role and function in glial cells as well as other cells within the CNS system, focusing the attention on their role in several neurodegenerative disease.

The work is comprehensive, with many lights and few shadows:

- among the lights are the extensive citation and an effort to cluster relevant articles published by thematic/functionality of the ABCs transporters,

- shadows include its somehow superficial breath since does not dig in depth on the molecular mechanisms proposed for their role in the original articles they review.

We included more information and revised fully the text. We also included a section in paragraph 3 about examples of molecular mechanisms where ABC transporters are involved which could have relevance for neurodegenerative diseases.

Nevertheless, I believe it constitutes a good starting point for all researches that begin to approach this rather complex field, to get handy and quick overview of the relevant literature of the field.

We fully revised the main text and the figures/figure legends to enable smooth reading. We also corrected the language style and did a proofreading for English. We added more information about molecular mechanisms and added a separate section in paragraph 3 to account for this needed information. Where appropriate, new information and citations were added.

Round 2

Reviewer 1 Report

Comments and Suggestions for Authors

I don't have any further comment.